# Identification and Characterization of Bacteria-Derived Antibiotics for the Biological Control of Pea Aphanomyces Root Rot

**DOI:** 10.3390/microorganisms10081596

**Published:** 2022-08-08

**Authors:** Xiao Lai, Dhirendra Niroula, Mary Burrows, Xiaogang Wu, Qing Yan

**Affiliations:** 1Department of Plant Sciences and Plant Pathology, Montana State University, Bozeman, MT 59717, USA; 2College of Agriculture, Guangxi University, Nanning 530004, China

**Keywords:** antibiotics, 2,4-DAPG, biocontrol, *Pseudomonas* spp., *Aphanomyces euteiches*

## Abstract

Antibiosis has been proposed to contribute to the beneficial bacteria-mediated biocontrol against pea Aphanomyces root rot caused by the oomycete pathogen *Aphanomyces euteiches*. However, the antibiotics required for disease suppression remain unknown. In this study, we found that the wild type strains of *Pseudomonas protegens* Pf-5 and *Pseudomonas fluorescens* 2P24, but not their mutants that lack 2,4-diacetylphloroglucinol, strongly inhibited *A. euteiches* on culture plates. Purified 2,4-diacetylphloroglucinol compound caused extensive hyphal branching and stunted hyphal growth of *A. euteiches*. Using a GFP-based transcriptional reporter assay, we found that expression of the 2,4-diacetylphloroglucinol biosynthesis gene *phlA*_Pf-5_ is activated by germinating pea seeds. The 2,4-diacetylphloroglucinol producing Pf-5 derivative, but not its 2,4-diacetylphloroglucinol non-producing mutant, reduced disease severity caused by *A. euteiches* on pea plants in greenhouse conditions. This is the first report that 2,4-diacetylphloroglucinol produced by strains of *Pseudomonas* species plays an important role in the biocontrol of pea Aphanomyces root rot.

## 1. Introduction

Biological control of soil-borne plant diseases by beneficial bacteria is a promising ecofriendly disease management strategy. Commercial application of beneficial bacteria-mediated biological control can be improved by increasing the disease control efficacy and reducing the control variations [1]. Understanding how beneficial bacteria suppress plant diseases will help us increase the efficacy and consistency of biological control [2].

Aphanomyces root rot of pea (*Pisum sativum*), caused by the oomycete pathogen *Aphanomyces euteiches*, is one of the most damaging soil-borne diseases that constrain pea production worldwide [3]. Interest in using beneficial bacteria to control *A. euteiches* is growing because of the unavailability of resistant pea cultivars, limited chemical management options, concerns of potential fungicide resistance by the pathogens, and the increasing requests for ecofriendly management alternatives [4,5]. Several beneficial bacteria, including strains of *Pseudomonas* spp., *Bacillus mycoides*, *Rhizobium* spp., *Pantoea agglomerans*, *Lysobacter capsici*, and *Burkholderia cepacia*, were found to inhibit the *A. euteiches* growth in culture media and/or reduce the disease incidence in greenhouse [6,7,8]. Results of these studies suggest that antibiosis likely contributes to the pathogen suppression and disease control. For example, the beneficial bacteria that reduced the disease severity of pea Aphanomyces root rot in growth camber biocontrol assays were also found to inhibit the *A. euteiches* mycelial growth on culture plates [8]. However, the antibiotic(s) required by the beneficial bacteria to inhibit the *A. euteiches* growth and control the disease remain(s) unknown. Considering the well-recognized and important roles of antibiotics in biocontrol [9], filling this knowledge gap will advance our understanding of the biocontrol mechanism of pea Aphanomyces root rot and improve the disease control efficacy of the antibiotic-producing beneficial bacteria.

The goals of this study were to identify beneficial bacteria-derived antibiotics that inhibit the growth of *A. euteiches* and characterize the role of those antibiotics in the tri-trophic interactions of *A. euteiches*, pea plant and the beneficial bacteria. Antagonistic bacterial isolates that belong to different genera, including *Pseudomonas*, *Paenibacillus*, *Bacillus*, *Pseudarthrobacter*, and *Chryseobacterium*, were identified from pea rhizosphere soils in this work. We focus on species of the *Pseudomonas* genus because strains of this group have been constantly identified to suppress pea Aphanomyces root rot [8,10]. *Pseudomonas protegens* strain Pf-5, a soil bacterium which is well-known for its wide-spectrum antimicrobial activity due to the production of many antibiotics including pyoluteorin, pyrrolnitrin, 2,4-diacetylphloroglucinol (2,4-DAPG), orfamide A, rhizoxin, hydrogen cyanide [11,12,13,14], and *Pseudomonas fluorescens* strain 2P24, a wheat rhizosphere bacterium with antibacterial and antifungal activities [15,16], were used in this study. The inhibitory effects of *P. protegens* Pf-5 and *P. fluorescens* 2P24 on the growth of *A. euteiches* were measured. The antibiotic required by *P. protegens* Pf-5 and *P. fluorescens* 2P24 to inhibit *A. euteiches* was identified and the roles of the antibiotic in biological control of pea Aphanomyces root rot were investigated.

## 2. Materials and Methods

### 2.1. Strains and Cultural Conditions

The oomycete and bacterial strains, plasmids, and sequences of oligonucleotides used in this study are listed in Table 1. Strains of *P. protegens* Pf-5 and *P. fluorescens* 2P24 were cultured at 28 °C on King’s medium B (KB) [17] or Nutrient Broth (Becton, Dickenson and Company, Franklin Lakes, NJ, USA) with or without agar (15%). Liquid cultures were grown with shaking at 250 rpm. *A. euteiches* was cultured at room temperature on half strength potato dextrose agar (½ PDA) (Becton, Dickenson and Company) plates that are commonly used to culture plant-associated oomycetes [18,19].

Phloroglucinol was purchased from Sigma-Aldrich (St. Louis, MO, USA), monoacetylphloroglucinol was purchased from Sigma-Aldrich Chemie GmbH (Schnelldorf, Germany), and 2,4-diacetylphloroglucinol was purchased from Toronto Research Chemicals (North York, ON, Canada). The compounds were dissolved in methanol.

### 2.2. Isolation and Identification of Pea-Associated Bacterial Antagonists

Rhizosphere soils were collected in June 2020 from lentil-growing fields at Valley County (GPS: 48°43.352′ N; 106°05.213′ W), Montana. The rhizosphere soils were suspended in 10 mL sterilized water and the suspensions were diluted onto ½ and ^1^/_10_ TSA (tryptone soy agar) plates that were incubated at 28 °C for 2–5 days. Bacterial isolates with diverse colony morphologies were collected from the plates and their inhibition against the growth of *A. euteiches* was tested as described below. The isolates that showed clear inhibition against *A. euteiches* were confirmed in a repeat experiment and stored at −80 °C in a freezer and their taxonomies were identified via 16s DNA analysis using oligonucleotide pairs 27f/1492r [20].

The DNA sequences of the PCR amplified 16s rRNA genes were determined by standard Sanger sequencing at GeneWiz (Azenta Life Sciences, Chelmsford, MA, USA), deposited into GenBank (accession numbers were shown in the phylogenetic tree in Figure 1C), and analyzed using BLAST with the NCBI nr/nt database to identify the closely related bacterial strains. The 16s rRNA sequences of the bacterial isolates that were identified in this study and the closely related strains that were retrieved from the NCBI database were aligned using a web-based sequence aligner program MAFFT v 7.490 [21]. The aligned sequences in the Phylip (.phy) format were then used for the phylogenetic analysis using CIPRES Science Gateway [22]. The phylogenetic tree was generated using the IQ-tree on XSEDE v 2.1.2 [23] based on the maximum likelihood approach. The GTR nucleotide substitution model was used, and the number of bootstrap iterations was set as 1000. The numbers at the tree nodes represent bootstrap support. The Figtree v 1.4.4 (http://tree.bio.ed.ac.uk/software/figtree) was used for the visualization of the phylogenetic tree. The scale at the bottom represents the number of nucleotide substitutions per site.

**Table 1 microorganisms-10-01596-t001:** Bacterial strains, plasmid, and primers used in this study.

Strains, Plasmids, and Primers	Genotypes, Relevant Characteristics, and DNA Sequences ^#^	Reference or Source
*A. euteiches*	Oomycete pathogen isolated from pea Aphanomyces root rot disease sample in Montana.	[24]
***P. protegens* strains:**		
LK099	Wild type Pf-5, soil isolate, DAPG^+^, Ofa^+^, Prn^+^, Plt^+^, HCN^+^, Rzx^+^.	[25]
JL4975	Δ*gacA*_Pf-5_, altered in the many phenotypes regulated by GacA, DAPG^−^, Ofa^−^, Prn^−^, Plt^−^, HCN^−^, Rzx^−^.	[11]
LK147	6-fold mutant of Pf-5, Δ*pltA*Δ*pcnC*Δ*ofaA*Δ*rzxB*Δ*hcnB*Δ*phlA*, DAPG^−^, Ofa^−^, Prn^−^, Plt^−^, HCN^−^, Rzx^−^.	[26]
LK107	5-fold mutant of Pf-5, Δ*pltA*Δ*pcnC*Δ*ofaA*Δ*rzxB*Δ*hcnB*, produces DAPG^+^, Ofa^−^, Prn^−^, Plt^−^, HCN^−^, Rzx^−^.	This study
LK023	Δ*phlA*_Pf-5_, DAPG^−^.	[27]
LK557	*phlA* repaired, DAPG^+^.	This study
LK410	Δ*phlF*_Pf-5_, overproduces DAPG, DAPG^++^	This study
JL4776	Δ*ofaA*_Pf-5_, Ofa^−^	[28]
JL4793	Δ*pcnC*_Pf-5_, Prn^−^	[11]
JL4805	Δ*pltA*_Pf-5_, Plt^−^	[11]
JL4808	Δ*rzxB*_Pf-5_, Rzx^−^	[11]
JL4809	Δ*hcnB*_Pf-5_, HCN^−^	[29]
** *P. fluorescens* ** **strains:**		
LK500	Wild type strain 2P24, wheat rhizosphere isolate, wild type, produce DAPG and HCN.	[30]
LK501	Δ*gacA*_2P24_, altered in the many phenotypes regulated by GacA, DAPG^−^, HCN^−^	[31]
LK506	Δ*phlACBD*_2P24_, DAPG^−^. This mutant also has a deleted *phlG* gene and lacks PhlG that degrades DAPG to MAPG.	[32]
**Plasmids**		
pEX18Tx	Gene replacement vector with MCS from pUC18, *sacB*^+^ Tc^r^	[33]
pEX18Tc-*phlA*	pEX18Tc containing wild-type *phlA*_Pf-5_ gene for repair of *phlA* mutation in the chromosome.	This study
pEX18Tc-*phlD*	pEX18Tc containing wild-type *phlD*_Pf-5_ gene for repair of *phlD* mutation in the chromosome.	This study
pEX18Tc-Δ*phlF*	pEX18Tc containing *phlF*_Pf-5_ with a 546-bp in-frame deletion.	This study
pPROBE-NT	pBBR1 containing a promoter-less *gfp*, Km^r^	[34]
p*phlA*_promoter_-*gfp*	pPROBE-NT containing the promoter of *phlA*_Pf-5_ fused with a promoter-less *gfp.* Km^r^	This study
p*phlA*_translation_-*gfp*	pPROBE-NT containing the promoter and the first seven codons of *phlA*_Pf-5_ fused with a promoter-less *gfp.* Km^r^	This study
**Primers**	oligonucleotide sequences (5′ to 3′) *	
phlA-F3	ATAGGATCCTTAAGGATTTCGATGGTGG	
phlA-R3	ATAGGTACCTGTTGCGGTTGATGGTGTCGGCG	
phlA_HindIII-F	GGACACAAGCTTCCCTATTTGGAGTCTGCTGT	
phlA_HindIII-R	CACACCAAGCTTTTCACATTCAGTGCTGGAGC	
gfp-phlA-f1	CTGCAGGTCGACTCTAGAGTCGATGGTGGAAGTGAGAATG	
gfp-phlA-r1	AGTGAAAAGTTCTTCTCCTTTACTCATGACAATACCTATCTTTTTCAC	
phlA-gfp-f1	GTGAAAAAGATAGGTATTGTCATGAGTAAAGGAGAAGAA CTTTTCAC	
phlA-gfp-r1	CATTCTCACTTCCACCATCGACTCTAGAGTCGACCTGCAG	
phlF-UpF-Hind3	TATAAGCTTGAGGTCGGTGTTTTTCC	
phlF-UpR-ovlp	TCAACGTTGCGTACCAGGACAAGAGCCGATGGAGCTGCG	
phlF-DnF-ovlp	CGCAGCTCCATCGGCTCTTGTCCTGGTACGCAACGTTGA	
phlF-DnR-EcoRI	ATAGAATTCAAGTGGTGGTTCATCTGG C	
Pf5-phlD-UpF-Hind	CGACACAAGCTTCAGTGCGAAGAATGCAACGA	
Pf5-phlD-DnR-Hind	CCTCTCAAGCTTTGGTGACAATGATGCTGGTG	
27f	AGAGTTTGATCCTGGCTCAG	
1492r	CGGTTACCTTGTTACGACTT	

^#^: Phenotype abbreviations: DAPG, 2,4-diacetylphloroglucinol; HCN, hydrogen cyanide; Ofa, orfamide A; Plt, pyoluteorin; Prn, pyrrolnitrin; Rzx, rhizoxin derivatives; abbreviations of antibiotics and their concentrations used in this work are: Tc, tetracycline (10 μg/mL for *E. coli*, 200 μg/mL for Pf-5); Km, kanamycin (50 μg/mL). *: underlines show DNA restriction enzyme sites that were used for cloning.

### 2.3. Inhibition Assays against A. euteiches

Inhibition assays were conducted on ½ PDA plates. A 5 mm agar plug from a 4-day old culture of *A. euteiches* was placed onto the center of the plates. Bacterial cells were washed by sterilized water to make a cell density at OD600 = 0.1. A drop of 3 μL of the bacterial cell suspension was placed on the agar surface about 2.5 cm away from the center of an *A. euteiches* plug. Inoculated plates were air-dried and incubated at 28 °C without light. Data were collected 2–3 days after inoculation by measuring the distance from the growing center of *A. euteiches* to the bacterial inoculation site. At least three replicates were used for each of the tested strains in one experiment. The experiment was repeated three times independently.

For the inhibition assay using purified compounds, the design was the same as above except the compounds were dissolved in methanol to make a 100 mM stock. A serial dilution was done using methanol to make appropriate concentrations. One μL of each concentration was added to a 6 mm-diameter filter paper disc and air-dried before placing it onto ½ PDA plates.

To observe the impact of 2,4-DAPG on the hyphal morphology of *A. euteiches*, the oomycete hyphae from the growing edge closest to the filter paper disc were collected immediately after the growth inhibition started and examined under a microscope. The experiment was repeated two times.

### 2.4. Construction of Pf-5 Mutant and Derivatives

The Δ*phlF*_Pf-5_ deletion mutant of Pf-5 was made by following our previous method [35]. Briefly, two DNA fragments flanking the *phlF*_Pf-5_ gene were PCR amplified from the genome of Pf-5 using two oligonucleotide pairs, phlF-UpF-Hind3/phlF-UpR-ovlp and phlF-DnF-ovlp/phlF-DnR-EcoRI (Table 1). These two DNA fragments were fused together by PCR and resulted in a 1204-bp DNA fragment that contains a 546-bp in-frame deletion of *phlF*_Pf-5_ gene. The PCR product was digested by *Hin*dIII and *Eco*RI and ligated into pEX18Tc to make a deletion construct pEX18Tc-Δ*phlF*. This construct was transferred into wild type Pf-5 to delete *phlF*_Pf-5_ gene in the chromosome. The deletion of *phlF*_Pf-5_ was confirmed by PCR and subsequent DNA sequencing.

The 5-fold mutant of Pf-5 (Δ*pltA*Δ*pcnC*Δ*ofaA*Δ*rzxB*Δ*hcnB*, strain ID LK107) which produces 2,4-DAPG but lacks pyrrolnitrin, orfamide A, rhizoxin, and hydrogen cyanide was made by introducing a wild type *phlD*_Pf-5_ gene to replace the deleted *phlD*_Pf-5_ in a previous made 6-fold mutant (Δ*pltA*Δ*phlD*Δ*pcnC*Δ*ofaA*Δ*rzxB*Δ*hcnB*, strain ID JL4909) [36]. Briefly, a 2103-bp DNA fragment containing the wild type *phlD*_Pf-5_ gene and its flanking sequences was PCR amplified from the genome of Pf-5 using oligonucleotide pairs Pf5-phlD-UpF-Hind/Pf5-phlD-DnR-Hind. The PCR product was digested by *Hin*dIII and ligated into pEX18Tc to make a complementation construct pEX18Tc-*phlD*. This construct was transferred into the 6-fold mutant (JL4909) to restore the *phlD*_Pf-5_ gene in the chromosome.

A similar strategy was used to replace the deleted *phlA*_Pf-5_ gene in the Δ*phlA*_Pf-5_ mutant’s chromosome with a wild type copy of the *phlA*_Pf-5_ gene. Briefly, a 1700-bp DNA fragment which contained the wild type *phlA*_Pf-5_ gene and its flanking sequences was PCR amplified using oligonucleotide pairs phlA_HindIII-F/phlA_HindIII-R, digested by *Hin*dIII, and ligated into pEX18Tc to make a complementation construct pEX18Tc-*phlA*. This construct was transferred into the Δ*phlA*_Pf-5_ mutant to restore the *phlA*_Pf-5_ gene in the chromosome.

### 2.5. Construction of Reporter Strains and GFP Activity Assays

The construction of GFP reporter constructs and the GFP activity assays were modified from our previous report [37]. Briefly, to measure the transcriptional expression of *phlA*_Pf-5_, a 1001-bp DNA fragment which contained the promoter of *phlA*_Pf-5_ gene was PCR amplified from the Pf-5 genome using oligonucleotide pair phlA-F3/phlA-R3 (Table 1). The PCR product was digested with *Bam*HI and *Kpn*I and ligated to pPROBE-NT to create the reporter construct p*phlA*_promoter_-*gfp*. To measure the translational expression of *phlA*_Pf-5_, a 570-bp DNA fragment which contained the promoter and the first seven codons of *phlA*_Pf-5_ was PCR amplified from the Pf-5 genome using oligonucleotide pair gfp-phlA-f1/gfp-phlA-r1 (Table 1). The vector pPROBE-NT was PCR amplified using primers phlA-gfp-f1/phlA-gfp-r1 and the PCR product was digested with *Kpn*I. The purified two DNA fragments were assembled using NEBuilder HiFi DNA Assembly Master Mix (NEB, catalog no. E2621L) to make p*phlA*_translation_-*gfp*. The reporter constructs were confirmed by sequencing analysis and transferred into the strain Pf-5.

To measure the GFP activity of Pf-5 reporter strains, wild type Pf-5 containing the above made construct p*phlA*_promoter_-*gfp*, p*phlA*_translation_-*gfp* or the empty vector were cultured overnight on KB plates plus kanamycin at 28 °C. Then the cultures were inoculated into NBGly plus kanamycin, a non-conducive medium for 2,4-DAPG production, at 28 °C with shaking from early morning until late afternoon. The bacterial cells were washed three times with sterilized distilled water and diluted to an optical density of 600 nm (OD_600_) equal to 0.02 in phosphate buffered saline (PBS). Field pea seeds (*P. sativum* L., cv. Carousel yellow) were surface sterilized twice with 3% sodium hypochlorite, 3 min per time, and rinsed three times with sterilized distilled water. The seeds were pregerminated for 3 days before use. Germinated pea seeds, five-day old *A. euteiches* plug (5 mm in diameter) and 5 mL of a bacterial suspension in PBS were added to test tubes (20 mm diameter opening). The test tubes were incubated at 28 °C without shaking. The OD_600_ was measured using a 96-well plate reader (SpectraMax M2) to monitor growth of bacteria in the PBS. The green fluorescence was monitored at 24 h by measuring emission at 535 nm with an excitation at 485 nm. Background GFP levels of the PBS solutions containing *A. euteiches* plugs or the germinated seeds were subtracted from the observed GFP activity of the bacterial reporter strains. The GFP levels of wild type Pf-5 with pPROBE-NT empty vector were further subtracted as another background correction. For each measurement, the GFP value was divided by the corresponding OD_600_ to determine the relative GFP level. The experiment included three replicates for each treatment and was repeated two times independently.

### 2.6. Biocontrol Assays and Disease Evaluations

Oospores of *A. euteiches* were prepared in homogenized oatmeal broth (5 g rolled oats per liter of water) using a previous method [38]. Mycelial mats in water were blended for 5 min and the oospore concentration was determined by a hemocytometer. The oospores were diluted in water and mixed with greenhouse soils (pH: 7.3; NO3-N: 68 ppm; PO4-P: 3 ppm; potassium: 47 ppm) to make a final concentration of 50 oospores per gram soil. Soils mixed with the same volume of water served as non-inoculated control. The mixed soils were evenly filled in 10-cm square pots. Field pea seeds (*P. sativum* L., cv., Carousel yellow) were surface sterilized twice with 3% sodium hypochlorite, 3 min per time, and rinsed three times with sterilized distilled water. The seeds were then air-dried in a biosafety cabinet. Pf-5 derivative strains used in greenhouse assays included the 5-fold mutant (LK107), which produces 2,4-DAPG but lacks pyoluteorin, pyrrolnitrin, orfamide A, rhizoxin, and hydrogen cyanide, and the 6-fold mutant (LK147), which lacks DAPG and the above five antibiotics. The bacterial strains were cultured on KB plates at 28 °C overnight. The cultures were transferred to KB liquid medium and incubated at 28 °C overnight with shaking at 250 rpm. Then the cells were washed with sterilized, distilled water three times and diluted to OD_600_ = 0.1. The seeds were treated with the bacterial suspension for three hours at room temperature with gentle shaking (around 60 rpm). Then, the seeds were removed from the suspension and planted in pots in the greenhouse. The planted pots were placed in a shallow tray and were watered the next day after planting by filling the tray with water. Greenhouse conditions were set to 22 °C day/18 °C night with a 16-h photoperiod. Each treatment contained six replicates and each replicate contained six seeds. The experiment was repeated two times.

The disease severity was evaluated by following a previous method [39]. Briefly, pea plants were harvested three weeks after planting. The roots were washed with tap water and examined for root rot symptoms. The disease severity scales were: 0 = healthy roots with no visible symptoms of root rot; 1 = slight water-soaking on the primary or secondary roots; 2 = moderate water-soaking on the primary or secondary roots or epicotyls with light-brown areas and more extensive discoloration; 3 = infected areas extensive and soft, but the entire root was not collapsed and the epicotyl was not markedly shriveled; 4 = extensive discoloration of the roots with tissue collapse and disintegration, or plant dead.

### 2.7. Statistical Analysis

The statistical analysis of the data collected in this study was performed by Student *t*-test or one-factor ANOVA analysis.

## 3. Results

### 3.1. Isolation and Identification of Bacterial Antagonists

Ten bacterial isolates that show clear antagonistic activities against *A. euteiches* on ½ PDA plates were identified from lentil field soil samples in this study (Figure 1). Among these ten isolates, four were identified as *Pseudomonas* spp., three were identified as *Paenibacillus* spp., and the other three were identified as *Bacillus* sp., *Pseudarthrobacter* sp. and *Chryseobacterium* sp., respectively, via 16s DNA sequencing and phylogenetic analysis (Figure 1C, Appendix A). Overall, the isolates of the *Pseudomonas* group showed stronger inhibition than strains of the other groups (Figure 1A,B).

**Figure 1 microorganisms-10-01596-f001:**
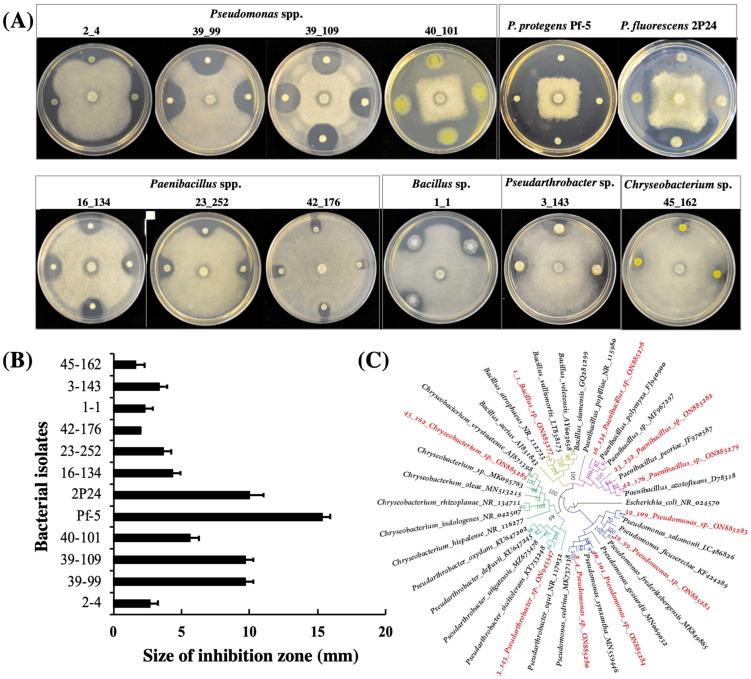
**Bacterial antagonists inhibit the growth of *A. euteiches*.***A. euteiches* was cultured in the center of the plates and the bacterial cells were inoculated around the pathogen. All strains were cultured on ½ PDA plates at 28 °C for three days before the results of the inhibition were recorded (**A**) and the size of the inhibition zone was measured (**B**). The experiment was repeated three times with similar results. (**C**); the taxonomy of the bacterial antagonists that were isolated in this work was identified by 16S rRNA analysis. The phylogenetic tree was constructed using IQ-tree and visualized using Figtree v 1.4.4. Five different bacterial genera forming distinct clades are represented by different colors at the branches. The ten bacterial isolates highlighted with red color were isolated in this study. The GenBank accession number of each isolate was shown. The values at the node of the tree represent bootstrap support.

We also included two well-studied *Pseudomonas* strains, *P. protegens* Pf-5 and *P. fluorescens* 2P24, in this work. Our results showed that both strains strongly inhibited the growth of *A. euteiches* on ½ PDA plates (Figure 1A,B).

### 3.2. 2,4-DAPG Biosynthesis Pathway Is Required for the Inhibition of A. euteiches by Strains Pf-5 and 2P24

We used *P. protegens* Pf-5 as a model to identify the antibiotic(s) that inhibit the growth of *A. euteiches*, because Pf-5 is known to produce many antibiotics including pyoluteorin, pyrrolnitrin, 2,4-diacetylphloroglucinol (2,4-DAPG), orfamide A, rhizoxin, and hydrogen cyanide [40]. We first tested a Pf-5 derivative, the Δ*gacA*_Pf-5_ mutant, which lacks the regulator protein GacA that globally controls the production of many secondary metabolites including the above-mentioned antibiotics [40]. Compared to the wild-type strain, the Δ*gacA*_Pf-5_ mutant has no clear inhibition of *A. euteiches* on the culture plates (0 mm of inhibition zone was recorded, Figure 2A), suggesting that GacA-controlled antibiotic production is required by Pf-5 to inhibit the pathogen.

In strain Pf-5, GacA positively controls the production of at least six different antibiotics including pyoluteorin, 2,4-DAPG, pyrrolnitrin, orfamide A, rhizoxin, and hydrogen cyanide. To test if these antibiotics are involved in the growth inhibition of *A. euteiches* by Pf-5, a 6-fold mutant which lacks all these six antibiotics was tested in the inhibition assays. The 6-fold mutant did not inhibit the pathogen’s growth (0 mm of inhibition zone was recorded, Figure 2B), indicating that at least one of these six antibiotics contributes to the growth inhibition against *A. euteiches* by Pf-5. We then tested six single mutants, each lacking the production of one of the six antibiotics due to a mutation of their biosynthesis genes (Figure 2C,D). Among these six mutants, five mutants that lack pyoluteorin, pyrrolnitrin, orfamide A, rhizoxin, or hydrogen cyanide inhibited the pathogen to a similar level (sizes of the inhibition zone were 9.1–10.4 mm) with the wild type strain (10.8 mm of the inhibition zones). The Δ*phlA*_Pf-5_ mutant, which has an abolished biosynthesis pathway of 2,4-DAPG due to a mutation of its biosynthesis gene *phlA*_Pf-5_ (Figure 3A), shows almost no inhibition against *A. euteiches* (0 mm of inhibition zone was recorded, Figure 2D). 2,4-DAPG is a polyketide antibiotic that is toxic to many plant pathogens including bacteria, fungi, oomycetes, and nematodes [41,42,43]. To confirm the role of the 2,4-DAPG pathway in the growth inhibition, we restored the Δ*phlA*_Pf-5_ mutant by replacing the deleted *phlA*_Pf-5_ gene with its wild-type copy in the chromosome. The repaired strain *phlA*^+^ inhibited *A. euteiches* to a similar level as the wild type Pf-5 (Figure 2B). Similarly, a 5-fold mutant, which has a restored 2,4-DAPG biosynthesis pathway in the 6-fold mutant background, strongly inhibited *A. euteiches*. Further, we tested a Δ*phlF*_Pf-5_ mutant, which overexpresses 2,4-DAPG biosynthesis genes due to a mutation of the transcriptional repressor PhlF (Figure 3A) that negatively regulates the expression of 2,4-DAPG biosynthesis genes [44]. Compared to the wild type, the Δ*phlF*_Pf-5_ mutant has a slightly stronger inhibition (12.0 mm of inhibition zone was recorded) against the growth of *A. euteiches* (Figure 2A). Taken together, these results indicate that the 2,4-DAPG biosynthesis pathway of Pf-5 contributes to the inhibition against *A. euteiches* on culture plates.

Strain *P. fluorescens* 2P24 is known to produce 2,4-DAPG and hydrogen cyanide [30]. Based on the results that the 2,4-DAPG biosynthesis pathway is required for Pf-5 to inhibit *A. euteiches*, we hypothesized that the same antibiotic pathway also contributes to the inhibition by strain 2P24. We tested this hypothesis by comparing the antagonistic activity of the wild type 2P24 and its two derivatives: a Δ*phlACBD*_2P24_, mutant, which lacks the four biosynthesis genes *phlACBD* for 2,4-DAPG production, and a Δ*gacA*_2P24_ mutant, which produces neither 2,4-DAPG nor hydrogen cyanide. Our results show that the Δ*phlACBD*_2P24_ and the Δ*gacA*_2P24_ mutants have similar levels of inhibition (0 mm of inhibition zones were recorded for both mutants), which were much decreased compared to the wild type strain (10.4 mm of inhibition zone was recorded) (Figure 2E). These results show that the 2,4-DAPG biosynthesis pathway contributes to the inhibition of *A. euteiches* by strain 2P24.

### 3.3. Purified 2,4-DAPG Suppresses A. euteiches Growth and Alters Its Mycelia Morphology

In *Pseudomonas* spp., the biosynthesis pathway of 2,4-DAPG generates three metabolites including two intermediates, phloroglucinol (PG) and monoacetylphloroglucinol (MAPG), and the final product 2,4-DAPG (Figure 3B). A previous study showed that the intermediates of the antibiotic pathway may play a role in the inhibition against the target pathogens [45]. The Δ*phlACBD*_2P24_ mutant lacks all these three metabolites due to the deletions of their respective biosynthesis genes (Figure 3A,B). The Δ*phlA*_Pf-5_ mutant lacks MAPG and 2,4-DAPG but produces the first-step intermediate PG because *phlD* remains functional. The result that Δ*phlA*_Pf-5_ mutant has an abolished inhibition to *A. euteiches* (Figure 2D) suggests that PG did not inhibit the pathogen.

We directly tested the toxicity of the three metabolites produced by the 2,4-DAPG biosynthesis pathway against *A. euteiches*. Growth of *A. euteiches* was inhibited by 25 nmol of 2,4-DAPG or 500 nmol of MAPG but was not inhibited by PG even at 1 μmol (Figure 3C). The sizes of inhibition zone caused by 25 nmol of 2,4-DAPG, 500 nmol of MAPG, and 1 μmol of PG were 2.9 mm, 0.8 mm, and 0 mm respectively. These data show that, among the three metabolites, 2,4-DAPG is the most potent antibiotic against *A. euteiches* and likely plays a major role in the pathogen inhibition by strains Pf-5 and 2P24.

We then investigated the impact of 2,4-DAPG on mycelial growth of *A. euteiches* under a microscope. Compared to the normal hyphal branching and growth of *A. euteiches* in the control treatment, the 2,4-DAPG-treated pathogen showed excessive branching of the hyphae and stunted growth of the branches (Figure 3D), which is consistent with the reduced mycelial growth caused by the purified antibiotic metabolite and the producing bacteria.

### 3.4. Expression of the 2,4-DAPG Biosynthesis Genes Is Induced by Germinating Pea Seeds

Antibiotic production of beneficial microorganisms can be strongly regulated by environmental factors, host plants, and target pathogens, which then influences plant disease control suppression. To investigate if the bacterial 2,4-DAPG biosynthesis is regulated by pea seeds and *A. euteiches*, we measured the expression of 2,4-DAPG biosynthesis genes in the pea-*Pseudomonas*-*A. euteiches* tri-trophic system. Strain *P. protegens* Pf-5 was used as a model of the 2,4-DAPG producing bacteria in this experiment. A reporter construct, p*phlA*_promoter_-*gfp*, which contains the promoter of *phlA*_Pf-5_ fused with a promoter-less *gfp*, was made to measure the transcription of the 2,4-DAPG biosynthesis genes. This reporter construct was transferred into the wild type Pf-5. The resultant reporter strain, Pf-5/p*phlA*_promoter_-*gfp*, was incubated with germinating pea seeds with or without *A. eut eiches*. As shown in Figure 4A, Pf-5/p*phlA*_promoter_-*gfp* had a significant higher GFP level when it was incubated with germinating pea seeds than with the buffer control. 

The presence of *A. euteiches* did not significantly influence *phlA* transcription by Pf-5, as assessed by the GFP level of the reporter strain. Similarly, the translational expression of *phlA*_Pf-5_, which was measured using the reporter strain Pf-5/p*phlA*_translation_-*gfp*, was also induced by the germinating seeds but not the pathogen (Figure 4B). These results indicate that expression of the 2,4-DAPG biosynthesis genes of Pf-5 is activated at both transcriptional and translational levels by germinating pea seeds which likely results in the production of 2,4-DAPG by the bacterium on pea seed surfaces.

### 3.5. 2,4-DAPG Plays a Role in the Biocontrol of Pea Aphanomyce Root Rot

Based on the above results that purified 2,4-DAPG and its producing bacteria strongly inhibited *A. euteiches* on culture plates, and that expression of 2,4-DAPG biosynthesis genes is activated by germinating pea seeds, we hypothesized that this antibiotic plays an important role in the biological control of *A. euteiches* root rot on pea. In the first step of testing this hypothesis, we set up an *A. euteiches*-pea root rot system and a biocontrol assay in greenhouse conditions. We found that inoculation of *A. euteiches* at a concentration of 50 oospores per gram soil caused a significant disease severity on the pea plants (Figure 5).

Next, we tested the disease control efficacy of the 5-fold mutant of Pf-5 which has a functional 2,4-DAPG biosynthesis pathway but lacks the other five antibiotics to avoid their potential impacts on plant development such as seed germination. Results show that pea plants that were treated by the 5-fold mutant have a significantly lower level of disease severity than the control plants treated by only the pathogen. Moreover, pea plants treated by the 6-fold mutant, which has a mutation in the 2,4-DAPG biosynthesis gene *phlA*_Pf-5_ in the background of the 5-fold mutant, developed a disease level that is significantly higher than the 5-fold mutant. Taken together, these results indicate that 2,4-DAPG plays an important role in the biocontrol of *A. euteiches* root rot by *P. protegens* on pea plants.

## 4. Discussion

In this study, we found that 2,4-DAPG produced by beneficial bacteria of the *Pseudomonas* genus plays an important role in the growth inhibition of the oomycete pathogen *A. euteiches* on culture plates and suppression of the pea Aphanomyces root rot disease in greenhouse. Our results are consistent with the previous reports that 2,4-DAPG produced by strains of *Pseudomonas* spp. is a major determinant in the biocontrol of many soilborne plant diseases due to its broad-spectrum toxicity against bacteria, fungi, oomycetes, and nematodes [41,42,43]. This is the first report that 2,4-DAPG inhibits *A. euteiches* and contributes to the beneficial bacteria-mediated biocontrol of pea Aphanomyces root rot disease.

Many beneficial bacteria produce multiple antibiotics that inhibit plant pathogens. Two beneficial bacterial strains, *P. protegens* Pf-5 and *P. fluorescens* 2P24, were used in this study to determine the role of antibiotics in the growth inhibition of *A. euteiches*. In addition to the 2,4-DAPG, strain 2P24 is known to produce another antibiotic hydrogen cyanide [30], and strain Pf-5 can produce at least another five antibiotics including pyoluteorin, pyrrolnitrin, orfamide A, rhizoxin, and hydrogen cyanide [40]. The important role of 2,4-DAPG in the inhibition of *A. euteiches* is clearly supported by the results that the Pf-5 and 2P24 mutants that lack 2,4-DAPG exhibited greatly reduced inhibition against *A. euteiches* compared to their wild type strains (Figure 2). The anti-oomycete activities of hydrogen cyanide, pyoluteorin, pyrrolnitrin, orfamide A, and rhizoxin have been reported previously in various phytopathogenic systems [46,47,48,49,50]. However, our results show that Pf-5 mutants that lack these five antibiotics retained strong inhibition against *A. euteiches*, suggesting that these five antibiotics of Pf-5 do not play a major role in the growth inhibition against *A. euteiches*. The important role of 2,4-DAPG in the inhibition of *A. euteiches* is also supported by the morphological changes in mycelia (excessive and stunted hyphal branches) of *A. euteiches* caused by purified 2,4-DAPG (Figure 3) and the reduced disease control efficacy of the 2,4-DAPG nonproducing mutant compared to the 2,4-DAPG producing derivative of Pf-5 (Figure 5). We recognized that the culture conditions used in the inhibition assays may not be conducive for the tested bacterial strains to produce all their antibiotics. For example, the ½ PDA medium used in this study contains a high level of glucose (2%, *w*/*v*) which likely represses the biosynthesis of pyoluteorin that is known to inhibit the oomycete pathogen *Pythium ultimum* [51]. Antibiotics other than 2,4-DAPG may contribute to the inhibition of *A. eutriches* by Pf-5 in different culture media.

Antibiotic production is critical for many beneficial bacteria to inhibit plant pathogens and suppress plant diseases, but expression of antibiotic biosynthesis genes is often influenced by the tri-trophic interactions between the beneficial bacterium, the pathogen, and the host plant [52,53,54]. Using a GFP-based transcriptional reporter system, we found that expression of the 2,4-DAPG biosynthesis gene *phlA*_Pf-5_ is strongly induced at both transcriptional and translational levels by germinating pea seeds (Figure 4). These results are consistent with our previous report that the 2,4-DAPG biosynthesis genes such as *phlD*_Pf-5_ are highly expressed in Pf-5 on the germinating pea seeds (*Pisum sativum* cv. Sugar Snap) in an RNAseq analysis and that 2,4-DAPG was detected on the Pf-5-treated pea seed surfaces [40]. These data support that production of 2,4-DAPG can be activated by the exudates of germinating pea seeds. Diverse metabolites including amino acids, carbohydrates, and organic acids were detected from pea seed exudates [40]. Our current understanding of how these exudates influence the antibiotic production of beneficial bacteria remains limited. Glucose is a known carbohydrate conducive to 2,4-DAPG production of Pf-5 [27,51] and has been detected in pea seed exudate at a moderate level (81.2 g/seed) [40]. However, the roles of the other more abundant carbohydrates including sucrose (1740.3 g/seed) and galactose (845.8 g/seed) and the most abundant amino acids including glutamate (206.4 g/seed) and arginine (86.1 g/seed) in the regulation of antibiotic production by Pf-5 remain unknown. Metabolites secreted by plant pathogens can also influence the antibiotic production of the associated beneficial bacteria. For example, fusaric acid, which is produced by *Fusarium* spp., inhibits 2,4-DAPG production of Pf-5 [36]. In this study, we found that the expression of *phlA*_Pf-5_ was not altered by the presence of *A. euteiches* hyphae (Figure 4). Infection of plant roots by pathogens can cause a metabolic leak that enhances antibiotic production of rhizosphere-associated beneficial microorganisms. For example, expression of *phlA* by the beneficial bacterium *P. protegens* CHA0 was increased on barley roots after infection by *Pythium ultimum* [55]. No or minimal damage of pea seeds were observed by the inoculated *A. euteiches* in our transcriptional reporter assays, probably due to the short incubation time (24 h) that was used to collect the GFP data of the bacterial reporter. It will be interesting to investigate the dynamic changes of the pea seed/root exudates, especially the metabolites that are known to regulate 2,4-DAPG production, at the different stages of the disease development of pea Aphanomyces root rot in the future.

In addition to 2,4-DAPG, many other antimicrobial compounds are known to be produced by strains of the *Pseudomonas* group and contribute to pathogen inhibition and disease biocontrol by the strains [56]. Characterizing different bacterial isolates that inhibit the growth of the pathogens can help us identify novel antibiotics to control the diseases. In this study, four *Pseudomonas* isolates were isolated from lentil filed soils and showed clear inhibition against *A. euteiches* (Figure 1). Among these four *Pseudomonas* isolates, 40_101 is interesting because of its strong inhibition which is comparable to the model strains Pf-5 and 2P24. The antibiotics produced by these strains were not characterized in this study. Future investigations, including whole genome sequencing analysis and bacterial mutagenesis, of these *Pseudomonas* isolates may identify new antibiotics for the disease manage of pea Aphanomyces root rot. Bacterial antagonists that belong to different genera including *Bacillus*, *Paenibacillus*, *Pseudarthrobacter*, and *Chreyseobacterium* were also identified in this work, although their inhibition effects are overall less than the *Pseudomonas* group. The antibiotic(s) that is(are) required for the growth inhibition of *A. euteiches* by these antagonists was (were) not investigated in this work. However, it is known that strains of *Bacillus* spp. and *Paenibacillus* spp. can produce diverse antibiotics such as lipopeptides that are toxic against plant pathogens including oomycetes [57,58]. The antagonistic bacteria were isolated from fields in Montana and have adapted to the local climates and soil/plant environments. Thus, they can serve as useful microbial resources for the development of biocontrol agents to manage pea Aphanomyces root rot disease in Montana.

Molecular identification using 16s rRNA is useful to identify bacteria although its limitation in accurate classification at species level has been recognized especially for strains of the *Pseudomonas* group [59,60]. The taxonomic classification of the ten bacterial strains isolated in this study was characterized to genus levels via 16s rRNA analysis (Figure 1) and can be further improved by multilocus sequence analysis or whole genome sequencing. Nevertheless, the result that *A. euteiches* was inhibited by bacterial isolates belonging to distinct genera suggests different antibiotics may be produced by these isolates to suppress the pathogen.

In conclusion, diverse groups of soil bacteria including *Pseudomonas* spp., *Bacillus* spp., *Paenibacillus* sp., *Pseudarthrobacter* sp., and *Chreyseobacterium* sp. were found to inhibit the growth of *A. euteiches*. Evidence obtained via genetic and biochemical analysis and GFP-based reporter assays supports that 2,4-DAPG is the major antibiotic required by model strains of the *Pseudomonas*, including *P*. *protegens* Pf-5 and *P. fluorescens* 2P24, to inhibit the growth of *A. euteiches* in culture medium and control the Aphanomyces root rot disease on pea plants. Similar approaches can be used in the future to identify and characterize antibiotics required by the isolated antagonistic bacteria to inhibit the growth of *A. euteiches*.

## Figures and Tables

**Figure 2 microorganisms-10-01596-f002:**
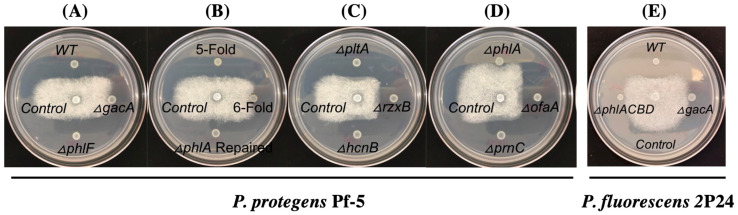
**Inhibition of Pf-5** (**A**–**D**) **and 2P24** (**E**) **and their derivatives against *A. euteiches* on plates.**
*A. euteiches* was cultured in the center of the plates and the bacterial cells were inoculated around the pathogen. WT: wild type; 5-fold: Pf-5 mutant, Δ*pltA*Δ*pcnC*Δ*ofaA*Δ*rzxB*Δ*hcnB*; 6-fold: Pf-5 mutant, Δ*pltA*Δ*pcnC*Δ*ofaA*Δ*rzxB*Δ*hcnB*Δ*phlA*; control has no bacteria inoculation. All strains were cultured on ½ PDA plates at 28 °C for two days before the results were recorded. Photos are representatives of three replicates and the experiment was repeated three times with similar results.

**Figure 3 microorganisms-10-01596-f003:**
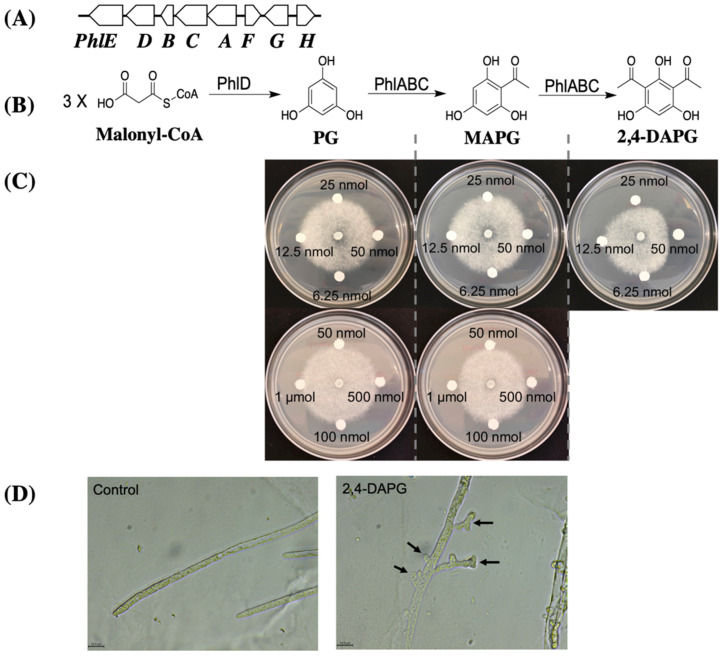
**Impact of metabolites generated in the 2,4-DAPG biosynthesis pathway on hyphal growth of *A. euteiches*.** (**A**), 2,4-DAPG gene cluster of Pf-5. Genes *phlABCD* encode four enzymes (PhlABCD) in the 2,4-DAPG biosynthesis pathway (**B**), and their expression is negatively regulated by the transcriptional repressor PhlF; PhlG degrades 2,4-DAPG to MAPG and its expression is controlled by PhlH; *phlE* encodes a putative permease of 2,4-DAPG. (**C**), Filter paper disks containing the indicated amount of the tested compounds were dried and placed around *A. euteiches* on ½ PDA plates. Representative results from three replicates were recorded two days after inoculation. (**D**), the oomycete hyphae were sampled at the *A. euteiches* growth margin on plates of (**C**) and examined under a microscope immediately after the inhibition was observed. Control indicates no compound treatment. Arrows show the excessive hyphal branching and stunted branches. Size bars indicate 15.9 μm. The experiments were repeated at least two times.

**Figure 4 microorganisms-10-01596-f004:**
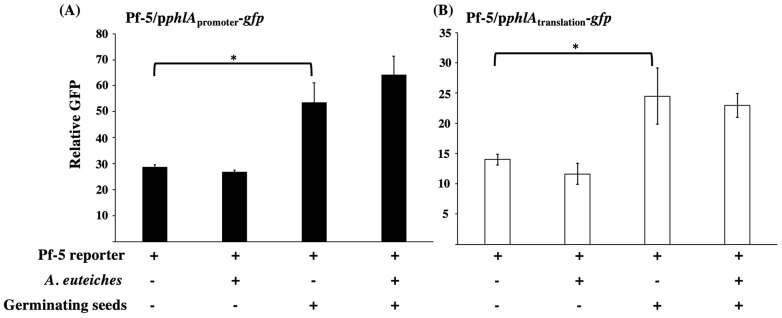
**Expression of *phlA* by Pf-5 in the interactions with pea germinating seeds and *A. euteiches*.** Wild type Pf-5 containing the GFP-based transcriptional reporter constructs p*phlA*_promoter_-*gfp* (**A**), or the translational reporter construct p*phlA*_translation_-*gfp* (**B**) was cultured with germinating pea seeds with or without the presence of *A. euteiches* hyphae. The expression of *phlA* was measured as relative GFP [fluorescence of GFP divided by (OD_600_ × 1000)] recorded at 24 h post inoculation. * indicates treatments are significantly different (*p* < 0.05), as determined by Student *t*-test. Data are means and standard deviations of three replicates from a representative experiment repeated two times with similar results.

**Figure 5 microorganisms-10-01596-f005:**
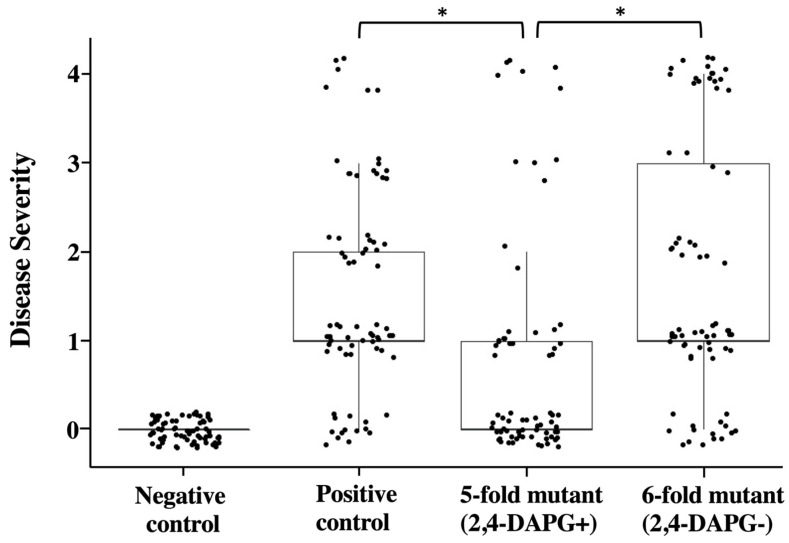
**Biocontrol assays of pea Aphanomyces root rot by *P. protegens* Pf-5 in greenhouse.** Pea seeds without inoculation served as negative control; pea seeds in positive control were planted in soil containing *A. euteiches* (50 oospores per gram soil) but not the Pf-5 strains. 5-fold mutant: Δ*pltA*Δ*pcnC*Δ*ofaA*Δ*rzxB*Δ*hcnB*; 6-fold mutant: Δ*pltA*Δ*pcnC*Δ*ofaA*Δ*rzxB*Δ*hcnB*Δ*phlA*. Disease severity was evaluated on a 0–4 disease level scale. Scatter plot shows combined data from two independent experiments each had six replicates (i.e., six growth pots) per treatment and each replicate included five to six pea plants. * indicates treatments are significant different (*p* < 0.05), as determined by one-factor ANOVA analysis.

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
