# Peer review of "Identification and Characterization of Bacteria-Derived Antibiotics for the Biological Control of Pea Aphanomyces Root Rot"

_microorganisms, 2022, doi:10.3390/microorganisms10081596_

Round 1
Reviewer 1 Report
Dear Editor,
Thank you once more for the chance to review this manuscript. While reading it again, I noticed that the authors have significantly improve the paper. Also, the authors have accepted all earlier comments and suggestions, so in my opinion the manuscript should be accepted as it is.
Author Response
Thank you.
Reviewer 2 Report
As I mentioned in my first review, it is an interesting paper, with a large array of in vitro and field tested results that make a good case. In addition, the paper is easy to read and the results were mostly clearly presented. The new additions of the paper have improved it. However, some things that I suggested in my first review concerning the 16S rRNA analysis were not properly discussed in this revised manuscript. I do not ask for additional phylogenomic analyses (that should be the standard nowadays for journals of an impact factor like Microorganisms, given the low expenses of bacterial NGS). However, the authors should not avoid acknowledging and discussing this rather weak point of the study.
Figure 1C. The phylogenetic tree is useful, however, the quality of the figure is low. In addition, I find it easier to read standard phylogenetic trees and not of the circular form. Although the tree shows the closest relative to each of the isolated strains, it does not necessarily mean that it is of that species. As I mentioned in my first review, the 16S rRNA has poor resolution at the species level. It is mostly good for genus identification. Maybe the tree could be a larger image of its own.
The 16S analysis is problematic in terms of identifying the bacterial species, especially for Pseudomonas fluorescens that is not one species, but a much wider taxonomic group. In addition, many species/strain misannotations have been observed in the Pseudomonas genus. For all the above, see https://doi.org/10.3390/d12080289. Better discuss them in the Discussion section.
The authors need to provide the best blast hit and sequence identity of each of the isolated strains too. Maybe I missed it, did the authors submit the 16S sequences to NCBI or some other Nucleotide sequence database?
Line 66-67: to the best of our knowledge.
Author Response
Thank you for the reviews. Please see our responses (in blue) below. The revised manuscript (supplementary files were not included due to restriction of the website) was attached for you to review.
As I mentioned in my first review, it is an interesting paper, with a large array of in vitro and field tested results that make a good case. In addition, the paper is easy to read and the results were mostly clearly presented. The new additions of the paper have improved it. However, some things that I suggested in my first review concerning the 16S rRNA analysis were not properly discussed in this revised manuscript. I do not ask for additional phylogenomic analyses (that should be the standard nowadays for journals of an impact factor like Microorganisms, given the low expenses of bacterial NGS). However, the authors should not avoid acknowledging and discussing this rather weak point of the study.
Response: A paragraph to acknowledge the limitation of using 16s rRNA analysis in bacteria identification was added in the revised manuscript.
Figure 1C. The phylogenetic tree is useful, however, the quality of the figure is low. In addition, I find it easier to read standard phylogenetic trees and not of the circular form. Although the tree shows the closest relative to each of the isolated strains, it does not necessarily mean that it is of that species. As I mentioned in my first review, the 16S rRNA has poor resolution at the species level. It is mostly good for genus identification. Maybe the tree could be a larger image of its own.
Response: Figure 1C was replaced by a larger image of the phylogenetic tree. A standard phylogenetic tree was added in supplementary file Figure S1.
The 16S analysis is problematic in terms of identifying the bacterial species, especially for Pseudomonas fluorescens that is not one species, but a much wider taxonomic group. In addition, many species/strain misannotations have been observed in the Pseudomonas genus. For all the above, see https://doi.org/10.3390/d12080289. Better discuss them in the Discussion section.
Response: agree, a paragraph to acknowledge the limitation of using 16s rRNA analysis in bacteria identification was added in the revised manuscript.
The authors need to provide the best blast hit and sequence identity of each of the isolated strains too. Maybe I missed it, did the authors submit the 16S sequences to NCBI or some other Nucleotide sequence database?
Response: the strains with the best blast hit were used in the phylogenetic analysis and the sequence identity between the bacterial isolates and their best blast hit strains from NCBI database was added in supplementary file Table S1. The accession numbers of the bacterial isolates were added in the Figure 1C, Figure S1 and Table S1.
Line 66-67: to the best of our knowledge.
Response: changed.

Reviewer 3 Report
The authors have modified and improved the manuscript taking into account the suggestions of the revferees. This version is suitable to be accepted for publication.
Author Response
Thank you.
This manuscript is a resubmission of an earlier submission. The following is a list of the peer review reports and author responses from that submission.
Round 1
Reviewer 1 Report
This is an interesting, rigorous and clearly written and exposed work, I only miss a small paragraph that allows at the end of the manuscript to gather at a glance the most relevant and novel conclusions.Reviewer 2 Report
Dear Editor,
Thank you for the opportunity to review the paper entitled "Identification and characterization of bacteria-derived antibiotics for the biological control of pea Aphanomyces root rot". In my opinion this paper must be significantly improved before any further steps in publishing process.
Some of my overall comments for this paper:
- Introduction part is very short and lacks of a certain concept that would explain what has already been done in the available literature. Also authors should emphasize the novelty of this work. Furthermore, in the introduction part that is related to goals of the study it is necessary to remove the conclusions, just to emphasize what is the aim of the study with underline novelty.
- Materials and Methods part should be supplemented with more information. For example:
line 67 – Why the ½ strength PDA and not fully PDA was used in this study. Please, provide adequate reference.
section 2.2. - the complete data about the isolation of bacterial antagonists are missing (time, temperature, site, geographical coordinates, relative humidity, etc.). Also, it is necessary to add which instrument was used for 16s DNA analysis.
line 169 – recepture of oatmeal broth is missing.
line 172 – Were the oospores dilute in tap or distilled water and why?
line 176 – Is there any proof that there is no residues of sodium hypochlorite in the sample after rinsing?
line 185 – please, provide exact parameter for gentle shaking
-Results part – presenting the most of the obtained results via Figures and not as determined parameters such as diameter of the inhibition zone is troublesome. Also, this study lacks in statistical approach which is mandatory for this type of experiments.
-Discussion part – discussion part is well-organized, but the main problem is the used literature. Namely, half of the references are older than a decade.
-Conclusion part – with the most important results and future perspectives of the study is completely missing
-References part – quite outdated. Total of 26 out of 46 references are older than 10 years.
Reviewer 3 Report
It is an interesting paper, with a large array of in vitro and field tested results that make a strong case. In addition, the paper is easy to read and the results were clearly presented.
Some suggestions/corrections:
In the introduction, the authors do not discuss why they chose to test 2,4-DAPG.
Line 368: that reference is rather old and the authors need to incorporate other more recent references on the role of 2,4-DAPG against fungi. For example, see doi: 10.1007/s10482-009-9407-7.
2,4-DAPG seems to affect the mitochondria and is fungistatic.
Line 80-81: The 16S analysis is problematic in terms of identifying the bacterial species, especially for Pseudomonas fluorescens that is not one species, but a much wider taxonomic group.
From figure 1, it is evident that the strains that the authors identified from the soil had moderate inhibition, apart from 40_101
Line 429-431: from figure 1 I understand that Bacillus and Paenibacillus mediated inhibition was rather poor, whereas the Pseudomonas spp. 40_101 demonstrated very good inhibition. Isn’t this rather inconsistent with this sentence in the Discussion? I would expect that the authors would discuss about this strain and their future plans. Probably sequence it? Its very easy and cheap nowadays. WGS would also place that strain in the correct position of the Pseudomonas genus phylogenomic tree.
The problem here is that the authors used 16S that has poor resolution. Pseudomonas is a very diverse genus, with many species/strain mis-annotations (see https://doi.org/10.3390/d12080289). In addition, P. fluorescens is a very diverse group, and not simply one species. It would benefit the paper, if the authors discussed all the above.
Line 4: An affiliation is missing
Line 49: belong to
Line 203-206: I guess the authors have identified these taxa with the 16S?
Line 216: required for the inhibition
Line 223: mentioned
Line 315: had a significantly higher
Line 322: in the production of
Line 375: at least another five
Line 410: please rephrase